# Directed Recovery and Molecular Characterization of Antibiotic Resistance Plasmids from Cheese Bacteria

**DOI:** 10.3390/ijms22157801

**Published:** 2021-07-21

**Authors:** Ana Belén Flórez, Lucía Vázquez, Javier Rodríguez, Baltasar Mayo

**Affiliations:** 1Departamento de Microbiología y Bioquímica, Instituto de Productos Lácteos de Asturias (IPLA), Consejo Superior de Investigaciones Científicas (CSIC), Paseo Río Linares s/n, 33300 Villaviciosa, Spain; lucia.vazquez@ipla.csic.es (L.V.); javier.rodriguez@ipla.csic.es (J.R.); baltasar.mayo@ipla.csic.es (B.M.); 2Instituto de Investigación Sanitaria del Principado de Asturias (ISPA), Avenida de Roma s/n, 33011 Oviedo, Spain

**Keywords:** antibiotic resistance, horizontal gene transfer, plasmids, tetracycline and erythromycin resistance, artisanal cheeses, lactic acid bacteria

## Abstract

Resistance to antimicrobials is a growing problem of worldwide concern. Plasmids are thought to be major drivers of antibiotic resistance spread. The present work reports a simple way to recover replicative plasmids conferring antibiotic resistance from the bacteria in cheese. Purified plasmid DNA from colonies grown in the presence of tetracycline and erythromycin was introduced into plasmid-free strains of *Lactococcus lactis*, *Lactiplantibacillus plantarum* and *Lacticaseibacillus casei*. Following antibiotic selection, the plasmids from resistant transformants were isolated, analyzed by restriction enzyme digestion, and sequenced. Seven patterns were obtained for the tetracycline-resistant colonies, five from *L. lactis*, and one each from the lactobacilli strains, as well as a single digestion profile for the erythromycin-resistant transformants obtained in *L. lactis*. Sequence analysis respectively identified *tet*(S) and *ermB* in the tetracycline- and erythromycin-resistance plasmids from *L. lactis*. No dedicated resistance genes were detected in plasmids conferring tetracycline resistance to *L. casei* and *L. plantarum*. The present results highlight the usefulness of the proposed methodology for isolating functional plasmids that confer antibiotic resistance to LAB species, widen our knowledge of antibiotic resistance in the bacteria that inhabit cheese, and emphasize the leading role of plasmids in the spread of resistance genes via the food chain.

## 1. Introduction

The wide use of antibiotics in human and veterinary medicine, agriculture and aquaculture has promoted the appearance and spread of resistance to antibiotics, compromising their therapeutic effectiveness [1]. Understanding the mechanisms involved in the transmission of resistance is vital if we are to control its spread [2]. Efforts to stop it are certainly needed in clinical settings, but are also vital across interconnected ecosystems involving livestock, food, food waste, water and sewage, etc. [3,4,5,6]. Paying attention to all of these as a whole is known as the One Health approach [7].

Antibiotic resistance genes (ARG) can spread via horizontal transfer between bacteria that share the same habitat. Among the different mobile elements that may be involved, plasmids are thought to be key players. Plasmids are extrachromosomal DNA molecules that recruit the host cell machinery for their replication and copy number control [8]. The metabolic cost of plasmid replication and maintenance can place an energetic burden on bacterial cells [9], but plasmids can also encode traits that confer advantages to the host. These are frequently associated with survival and adaptation to changeable environments, including the ability to metabolize different carbon and nitrogen sources, resistance and tolerance to heavy metals, disinfectants, antibiotics, and other environmental pollutants, and the ability to synthesize antimicrobial agents [10,11,12,13]. By transferring beneficial traits within and between bacterial species, plasmids play an important role in these species’ evolution [10]. The transfer of plasmids between bacteria can occur via conjugation, transduction, transformation, and vesiduction [14]. The presence of ARG-carrying plasmids, therefore, increases the risk of the transfer of resistance [15,16]; the study of such plasmids is crucial if the risk they pose to human health is to be understood, and if strategies to reduce the horizontal transfer of the genes they carry are to be developed [12].

The food chain may be a major thoroughfare for the spread of antibiotic resistance from animals to humans, particularly by consumption of fresh, little-processed, and raw-made products [17,18,19]. In cheese, ARG have been identified in total microbial DNA as well as in isolated bacteria. Genes coding for resistance to tetracycline, erythromycin, streptomycin, aminoglycosides, chloramphenicol and virginiamycin, have all been reported [20,21], and Southern blotting and genome sequencing have shown many of these genes to be carried by plasmids [22,23,24]. Both narrow and broad host-range ARG-carrying plasmids have been frequently detected in Gram-negative bacteria isolated from food [25,26]. However, the abundance, diversity, and transfer capacity of ARG-carrying plasmids in Gram-positive bacteria, and more specifically in lactic acid bacteria (LAB; the majority microbial types in cheese), has been left largely unexplored. The plasmid complement of most LAB species is complex, making the detection of associations between plasmids and antibiotic resistance challenging and laborious [27,28], and certainly, high-throughput sequencing and assembly technologies have so far failed to associate antibiotic resistance genes with plasmids in genomic and metagenomics studies [29]. In this context, this work reports a simple method for isolating functional plasmids that confer antibiotic resistance to LAB species. The results provide knowledge on the prevalence and diversity of plasmids conferring tetracycline and erythromycin resistance to cheese bacteria, and provide preliminary insight into the role of plasmids in the spread of antibiotic resistance throughout dairy ecosystems.

## 2. Results

### 2.1. Diversity of Plasmid Profiles

Counts of resistant aerobic mesophilic bacteria and LAB were performed on Plate Count Milk (PCM) and de Man, Rogosa, and Sharpe (MRS) agar plates, respectively, both supplemented with tetracycline or erythromycin, from samples of Cabrales cheese at day 3 (3D; early manufacture) and day 60 (60D; end of ripening). Large counts of tetracycline- and erythromycin-resistant bacteria were enumerated in all samples analyzed; these were almost identical at the same sampling point. The values obtained in PCM plates ranged from 10^8^ to 10^6^ ufc mL^−1^ at day 3 and 60, respectively. However, the level of resistant bacteria in MRS plates were lower than in PCM plates, since in the former were observed values from 10^5^ ufc mL^−1^ during manufacture and 10^6^ ufc mL^−1^ at the end of ripening. Total plasmid DNA was isolated from semiconfluent colonies of total aerobic mesophilic bacteria (PCM-3D and PCM-60D) and LAB (MRS-3D and MRS-60D) grown in the counting plates. For a better comparison of the diversity, total plasmid DNA was initially digested using restriction enzymes before visualization by gel electrophoresis. Complex patterns of intense and weak plasmid bands were observed for all samples. Appendix A shows an example of the PstI and XhoI digestion patterns for the total plasmid DNA recovered from the different samples. Although the digestion profiles obtained on PCM and MRS at 3D sample look rather different, the number, intensity and size of the bands were similar at 60D sample.

### 2.2. Screening of Plasmids Providing Antibiotic Resistance

Plasmid DNA from the PCM-3D, PCM-60D, MRS-3D and MRS-60D samples was independently transformed into electrocompetent cells of *Lactococcus lactis* NZ9000, *Lactiplantibacillus plantarum* NC8, and *Lacticaseibacillus casei* LB23. Tetracycline- and erythromycin-resistant transformants were then selected on GM17 and MRS solid media supplemented with the required antibiotic. Overall, 54 tetracycline-resistant colonies of lactococci were obtained with the plasmid DNA from the PCM-3D (30 colonies) and MRS-3D plates (24 colonies), while 13 resistant colonies were obtained using plasmid DNA from the PCM-60D (8 colonies) and MRS-60D plates (5 colonies). Similarly, 10 and 15 tetracycline-resistant colonies of *L. casei* and *L. plantarum*, respectively, were found; these were all obtained with plasmid DNA from the PCM-3D sample. Finally, only five erythromycin-resistant colonies were recovered; these were obtained in *L. lactis* when using again plasmid DNA from the PCM-3D sample.

All resistant transformants were grown overnight in liquid media with antibiotics and analyzed for the presence of plasmid DNA; after extraction and purification, plasmids were detected in all 97 resistant colonies. Restriction analysis of plasmid DNA identified seven digestion profiles among the tetracycline-resistant transformants, five among the *L. lactis*, one each for the two *Lactobacillus* strains, and a single digestion profile among the erythromycin-resistant *L. lactis* cells (Figure 1). This suggested the presence of an equivalent number of different plasmid molecules, designated pTC1 to pTC7 for those providing tetracycline resistance, and pERM1 for that providing erythromycin resistance. pTC1 was the most widespread plasmid (found in 49 out of 92 tetracycline-resistant colonies). It was recovered in *L. lactis* with plasmid DNA from all four tetracycline-grown bacteria samples. The plasmid profile of pTC1 was the only one observed from the PCM-3D, PCM-60D and MRS-60D samples. In addition, unique plasmid digestion profiles were also detected among erythromycin-resistant colonies in lactococci (pERM1) and tetracycline-resistance colonies in lactobacilli strains (pTC6 in *L. casei* and pTC7 in *L. plantarum*). In contrast, five different plasmid profiles were obtained in *Lactococcus* with plasmid DNA from the MRS-3D sample. These were detected with variable frequency among the colonies analyzed: pTC1 (in 6 colonies), pTC2 (in 3), pTC3 (in 1), pTC4 (in 2), and pTC5 (in 12). The estimated size of the plasmids ranged from about 4 (pTC6) to 50 (pEMR1) kbp.

### 2.3. Antibiotic Susceptibility of Plasmid-Harboring Transformants

Transformant strains carrying the plasmids from pTC1 to pTC7 and pERM1 (representative plasmids of all different profiles) were tested for antibiotic susceptibility against a set of 16 antibiotics using a broth microdilution method (Sensititre) and the evaluator strip (MICE) system and the results compared to those for the untransformed parental strains. The results obtained are summarized in Table 1. *L. lactis* cells carrying tetracycline resistance plasmids showed a MIC for this antibiotic of 128–192 µg mL^−1^ compared to 1 µg mL^−1^ of the host strain *L. lactis* NZ9000. Transformants harboring pTC3 and pTC5 also showed increased MICs to streptomycin (32–64 versus 16 µg mL^−1^) and chloramphenicol (16–32 versus 4 µg mL^−1^). The presence of plasmids in *L. casei* and *L. plantarum* moderately increased the MIC to tetracycline, while MIC values for other antibiotics were unaffected (Table 1). Finally, the erythromycin-resistant *L. lactis* transformants showed a MIC of erythromycin >256 µg mL^−1^. Additionally, they showed increased MICs of streptomycin (>256 µg mL^−1^), neomycin (32 µg mL^−1^), tetracycline (>256 µg mL^−1^), clindamycin (>256 µg mL^−1^), and quinupristin-dalfopristin (>8 µg mL^−1^) (Table 1).

### 2.4. Detection of Tetracycline and Erythromycin Resistance Genes

The presence of tetracycline and erythromycin resistance genes in the plasmids responsible for the increased MICs to these antibiotics in host strains was initially evaluated by conventional PCR. Genes coding for ribosomal protection proteins conferring tetracycline resistance were amplified by PCR using the universal pair of primers DI-DII, and plasmid DNA from the tetracycline-resistant *L. lactis* transformants as a template. Subsequent gene-specific PCR analysis for the tetracycline resistance genes *tet*(W), *tet*(M), *tet*(S), *tet*(O), *tet*(K), and *tet*(L) showed positive amplification only when targeting *tet*(S) gene. This result was further confirmed by sequencing of the amplicons. Surprisingly, no amplification was ever obtained when DNA of the tetracycline-resistant lactobacilli transformants was used as a template. While the presence of several genes conferring erythromycin resistance (*ermA*, *ermB*, *ermC*, *ermF*, and *mefA*) were assessed, positive amplification was just obtained when using gene-specific primers for *ermB* and plasmid DNA from erythromycin-resistant *L. lactis* as a template. This gene codes for a methyltransferase which confers resistance to macrolides, lincosamides and streptogramins (the MLS phenotype), which might thus explain the high MIC value for clindamycin.

### 2.5. Sequencing, Assembly and Annotation of Plasmids

Representative plasmids of the different digestion patterns were selected for entire sequencing. High quality reads were assembled, annotated, and analyzed. After assembly, a single plasmid molecule was detected for all antibiotic-resistant plasmid patterns except for that of pTC6, the profile of which consisted of two plasmid molecules (pTC6.1 and pTC6.2). The overall G+C content of the plasmids ranged from 30.6% to 34.3% for those recovered in *L. lactis*, and from 34.3% to 39.5% for those in lactobacilli species. Appendix A shows the ORFs identified in each of the plasmids and the putative biological function of their deduced proteins. The majority of ORFs showed high homology with genes located on the plasmids or the chromosome of the *Lactococcus* species. However, some ORFs showed homology to genes found in other *Lactobacillales* species, such as *Streptococcus parauberis*, *Enterococcus faecalis*, *Enterococcus faecium*, etc. Further, a minority of the deduced amino acid sequences shared homology with proteins from *Listeria monocytogenes*, *Staphylococcus aureus*, and *Bacillus licheniformis*, such as (respectively) an ATP-dependent helicase (pTC4-ORF23 and pTC5-ORF22; Appendix A), a replication initiation factor domain-containing protein (pTC5-ORF 30; Appendix A), and a hypothetical protein (pTC1-ORF12; Appendix A). Additionally, a region of pERM1 encompassing ORF3, ORF4 and ORF5 showed homology (respectively) to genes from *Tissierella pigra*, *Halanaerobiaceae bacterium* and *Alkalibaculum bacchi*.

### 2.6. Tetracycline Resistance Plasmids

Nucleotide sequence analysis identified two modules in pTC1. The first covered 88% of the molecule and proved to be very similar (98% nucleotide identity) to a plasmid region of *S. parauberis* SPOF3K (CP025421.1), which also harbors a Tet(S)-encoding gene. Besides *tet*(S), this module accommodates genes coding for two replication proteins of the RepB family, and four ORFs encoding proteins involved in conjugation and mobilization (Figure 2; Appendix A). The second module showed high homology (98% nucleotide identity) to segments of plasmids such as, among others, pLd10 (MG813924.1), pUL8B (CP016705.1) and pAH82 (AF243383.1) of *L. lactis* subsp. *lactis*, and pNZ4000 (AF036485.2) and pJM3A (CP016737.1) of *L. lactis* subsp. *cremoris*.

A major part of pTC2 (ΔORF2 to ORF11) was identical to the first module of pTC1; this includes the *tet*(S) gene and the region encoding replication and mobilization functions. The pTC1 and pTC2 are of about the same size and differ only in a few genes encoding putative recombinases and hypothetical proteins (Figure 2; Appendix A).

pTC3 was organized into three different modules. The whole plasmid shares (with minor rearrangements) high homology with the entire sequence of the lactococcal plasmids p158F (CP016690.1) and pHP003 (AF247159.1). Beyond *tet*(S), the plasmid contained genes associated with chloramphenicol (*cat*; ORF26) and streptomycin (*str*; ORF31) resistance. The region containing the three antibiotic resistance genes is highly homologous to a segment flanked by two IS*6*-like transposases of pK214 (X92946.1). pTC3 also contained ORFs encoding three complete and two partial replication proteins of the RepB and RepC families and eight proteins associated with mobilization (Figure 2; Appendix A).

At 42,318 bp, pTC4 was the largest plasmid recovered from the tetracycline-resistant transformants. Structurally, it was organized as an apparent mosaic resulting from the fusion of segments of a variety of *L. lactis* plasmids, including fragments from pK214, pS127 (CP061323.1), p001F (CP053672.2), pLP712 (FJ649478.1), pC41 (AP018500.1), and pA12-4 (LT599053.1). Some regions of pTC4 showed strong identity to others from plasmids of *Lactococcus garvieae* (pNUF18; LC316979), *Macrococcus cannis* (pKM0218; MF477836.1), *S. aureus* (plasmid I; LT799381.1), *Carnobacterium divergens* (pMFPA43A1505B; LT984412.1), and *E. faecium* (e.g., pV24-4 [CP036155.1] and pA6521_3 [CP061820.1]). This agrees well with the presence of seven IS-like elements in the pTC4 molecule (Figure 2; Appendix A). Two such IS flanked the region harboring *tet*(S). Among other accessory genes, ORFs coding for the synthesis and secretion of a bacteriocin of the lactococcin family were also noted (Appendix A).

Much of the pTC5 sequence (77%), including its *tet*(S)-associated gene, was shown to be almost identical (99%) to that of pTC4. The differential region in pTC5 (from ORF26 to ORF37) was flanked by two transposases and comprised genes involved in resistance to streptomycin and chloramphenicol (Figure 2; Appendix A). This antibiotic resistance-dedicated region was very similar to a section of pTC3.

pTC6.1 and pTC6.2 showed 100% nucleotide identity to, respectively, pWCFS101 (CR377165.1) and pWCFS102 (CR377164.1) from *L. plantarum* WCFS1. These plasmids replicate by a rolling circle mechanism, in agreement with their respective prototypes, pC194 and pMV158, to which they bear homology. Apart from genes coding for replication-associated proteins, only ORFs coding for hypothetical proteins were detected. No tetracycline resistance determinants were identified (Figure 3, Appendix A).

Finally, the sequence of pTC7 showed only a five-nucleotide difference to that of plasmid p256 from *L. plantarum* DSM 20174 (AJ62894.1). It is worth noting the presence in pTC7 of ORFs encoding proteins involved in plasmid maintenance (ORF12 and ORF13), but not in replication initiation functions. Again as in pTC6.1 and pTC6.2, ORFs coding for dedicated proteins involved in tetracycline resistance were absent in pTC7 (Figure 3; Appendix A).

### 2.7. Erythromycin Resistance Plasmids

pERM1, the single plasmid recovered from the erythromycin-resistant *L. lactis* transformants, was organized into several modules bound by IS elements (Figure 4; Appendix A). Besides *ermB*, a functional *tet*(S) was also identified in the pERM1 sequence. The module encoding *ermB* (ORF33) and *tet*(S) (ORF29) genes showed strong nucleotide homology to a conserved set of structures identified in the chromosome of *Mammaliicoccus sciuri* GDK8D6P (CP065792.1) and *Streptococcus dysgalactiae* NTUH_1743 (EF682209.1), as well as in plasmids of *Macrococcus canis* (pKM0218; MF477836.1*), Lactococcus raffinolactis* (pLraf_19_4S_1; CP050535.1), *C. divergens* (pMFPA43A1405B; LT984412.1), and *L. lactis* (pUC08B; CP016727.1). In contrast, the module harboring other resistance genes, namely streptomycin (*str*), quinupristin-dalfopristin (*Vgb*), tunicamycin (*tmrB*), and aminoglycosides (*ant*[6]-Ia and *vat*), showed no significant homology with sequences deposited in the NCBI database. The remaining modules, which include ORFs coding for proteins involved in replication and conjugal transfer, proved to be identical to others from pS127 (CP061323.1) and pMRC01 (AE001272.1) in *L. lactis*.

### 2.8. Nucleotide Sequence of tet(S) and Its Flanking Regions

All six plasmids carrying the *tet*(S) gene (pTC1 to pTC5 and pERM1) shared an identical block of 3062 bp (Figure 5). This core region started 988 bp upstream of the ATG position of *tet*(S) and ended 133 bp downstream of its stop codon. Within these blocks, a single nucleotide transversion (C to A) in the sequence of pTC1 was noted, and a single nucleotide transition (C to T) in pERM1 (Figure 5). The *tet*(S) gene contains 1941 nucleotides with the capacity to encode a protein 646 amino acids long; the Tet(S) protein coded by all these plasmids was identical.

## 3. Discussion

Plasmids enable bacteria to recruit accessory traits, including resistance to antibiotics, that confer selective advantages in terms of colonization, development or persistence in different environments [10]. Plasmids, and thus resistances, can be spread via horizontal gene transfer events such as conjugation and mobilization to neighboring partners, causing real risks to human health if they reach pathogenic or opportunistic bacteria [30]. The presence of antibiotic resistance determinants in plasmids carried by food-associated bacteria are thought to pose the greatest risk of resistance spreading along the food chain [17].

Resistant strains and resistance genes can be detected via phenotypic and genetic surveys, respectively [31,32]. Some resistance genes have been identified and located to plasmids [24], and even isolated and characterized [33,34]. However, the assignment of resistances to plasmids can be challenging and laborious, requiring studies on transference and mobilization by conjugation [26,35,36,37], hybridization analysis [24,36,38], and the location of resistance genes by analysing assembled plasmids sequences [39,40,41]. Furthermore, currently-in-use high-throughput sequencing and assembly technologies have failed to associate antibiotic resistance genes with plasmids in genomic and metagenomics studies [29]. The present work, however, reports a simple method of retrieving plasmids able to replicate in specific hosts and capable of conferring resistance to a particular antibiotic. These plasmids can then be fully characterized at the phenotypic and genetic levels.

Although ARG-carrying plasmids might be extracted and purified from bacteria as they are naturally found in cheese, the present work involved preliminary enrichment with antibiotics to provide proof of concept. Tetracycline and erythromycin were selected due to their extensive use as therapeutic agents and as growth promoters in Europe until banned in 2006 [42]. Further, resistance to these two antibiotics has already spread in dairy environments [21,43] and certainly among LAB species [44,45].

Overall, five plasmids replicating in *L. lactis* ranging in size from 11 to 42 kb were found to confer resistance to tetracycline. Although *tet*(S), tet(M), *tet*(O), *tet*(W), *tet*(L), and *tet*(K) have previously been detected in dairy LAB [21,22,43,46], only *tet*(S) was detected in the tetracycline-resistant plasmids characterized in the present study. In a previous functional metagenomics study, *tet*(A), *tet*(M), *tet*(S) and *tet*(L) were found throughout Cabrales manufacture and ripening [22]. The presence of distinct tetracycline resistance genes in different cheese batches can be explained by the great heterogeneity of the microbial populations in cheeses made from raw-milk. A larger copy number of the plasmids carrying *tet*(S) —a possibility not assessed in the present study— might help mask other tetracycline resistance genes. Despite finding only *tet*(S), at least five different *L. lactis* plasmids carrying the gene were detected, suggesting the wide spread of *tet*(S) across lactococcal plasmids. In contrast, a single plasmid encoding erythromycin resistance was retrieved in the same species, which suggests *ermB* to be less abundant than *tet*(S) in cheese.

Sequence analysis revealed the presence of RepB family proteins in all *L. lactis* plasmids, suggesting they follow a theta-type replication mechanism [47], their replicons belonging to different theta-replicating plasmid families. Although no genes encoding replication proteins were detected in pTC7 from *L. plantarum*, a theta-type mode of replication has been proposed for its closest relative, p256 [48]. Plasmids with such a mode of replication are more stable than those that use the rolling circle mechanism. Further, large numbers of theta-replicating plasmids can coexist in a single cell [28,49,50,51], which agrees well with the high prevalence of these replicons in cheese bacteria. Nevertheless, theta-replicating plasmids have a narrow host range, sometimes a single species or a few of close relative species [52]. In agreement with this, in this work, the same plasmid DNA pool was electroporated into three species, but plasmids carrying dedicated resistance genes were only recovered from *L. lactis*, the species that allows replication. The same plasmids, and thus the same resistance genes, were retrieved from samples on day 3 and day 60, even though the bacterial populations and their associated resistances were different, see [22,53]. The likely existence of resistance plasmids in other cheese-borne bacteria demands other types of bacterial species be used as a host, including *Enterococcus* spp., *Staphylococcus* spp., *Streptococcus* spp., and *Escherichia coli*, all of which have been reported important in the spread of antibiotic resistance in cheese [22,53,54,55,56].

Numerous IS- and transposase-related sequences were identified delimitating the different modules of the plasmids, or even flanking distinct functional blocks within the modules. A mosaic plasmid structure bound by IS has been recognized as paramount for maintaining the plasticity necessary for LAB to adapt to the dairy environment [57,58]. This type of structure may help alleviate the energetic costs of hosting large plasmids and complex extrachromosomal elements [59,60]. The modular structure argues in favor of a common pool of exchangeable blocks, which could offer bacteria a changing plasmid complement, even under similar genetic backgrounds [61]. In this respect, the blocks harboring *tet*(S) proved to be almost identical to regions of the *S. parauberis* plasmid SPOF3 [62] and of pK214 from *L. lactis* K214 [63]. Indeed, the block harboring the chloramphenicol and streptomycin resistance genes was identical to a region of pK214 [63]. Similarly, the block encoding *ermB* was very similar to another in *M. canis* pKM0218 [64]. The surrounding regions of *tet*(S) were also identical in pTC1 through pTC5 and pERM1, which strongly suggests a recent horizontal transfer from a common source. The presence of such blocks in different plasmids, and in plasmids from different species and genera, further suggests that these blocks are the real spreadable elements, with plasmids the preferred host in *L. lactis* [2,9,11,28,58].

Increased MICs for some antibiotics have largely been attributed to non-specific mechanisms such as reduced antibiotic uptake, reduced cell permeability, the thickness and compactness of the cell wall, defective cell wall autolytic systems, and the presence of multidrug resistance transporters [65,66]. Although none of the proteins encoded by ORFs from pTC6.1-pTC6.2, or pTC7 could be assigned any of these functions, it cannot be ruled out that the moderate tetracycline resistance provided to *L. casei* and *L. plantarum* is associated with one or more of the above mechanisms.

Selection for tetracycline and erythromycin resistance was pursued, but genes conferring resistance to other antibiotics were also identified, namely streptomycin, quinupristin-dalfopristin, tunicamycin, chloramphenicol, and aminoglycosides. These genes, either on plasmids or in the chromosome, have been repeatedly detected in LAB species and strains [23,63,67,68]. Linkage in the same chloramphenicol and streptomycin resistance cluster might allow co-selection to explain the maintenance of resistance to these antibiotics, and possibly to others [69]. The linkage and ensuing co-selection of antibiotic resistance and adaptive biological functions has been hypothesized by many authors [59,70,71,72,73]. However, in the present work, no feature that would help LAB develop and compete in the milk environment (lactose utilization, proteinase activity, phage resistance, etc.) was envisioned in any of the resistance plasmids. The maintenance and transfer of resistance plasmids in the dairy environment appears, thus, to be promoted by the presence of antibiotic residues in milk [74,75,76], although it might also be explained by plasmid-encoded addictive mechanisms [77] or host chromosomal adaptations reducing any associated biological costs, resulting in no selection against a plasmid’s loss [58,78,79].

## 4. Materials and Methods

### 4.1. Lactic Acid Bacteria Strains and Culture Conditions

*L. lactis* NZ9000 was grown under aerobic conditions in M17 medium (Biokar, Beauvais, France) supplemented with 1% glucose (GM17) at 32 °C for 24–48 h. *L. plantarum* NC8 and *L. casei* BL23 were cultured in de Man, Rogosa, Sharpe medium (MRS) (Merck, Darmstadt, Germany) at 32 °C for 24–48 h. Agar plates were obtained by supplementing the respective broth media with 2% agar (Merk).

### 4.2. Cheese Sampling

Samples of Cabrales cheese were taken during the manufacturing (3 days) and ripening (60 days) stages of production, homogenized in a Stomacher (Seward, Worthing, UK) with 2% (*w*/*v*) sodium citrate (Merck), and serially diluted in Ringer’s solution (Merck). Aerobic mesophilic bacteria and LAB resistant to tetracycline and erythromycin were enumerated by plating the dilutions on Plate Count Milk (PCM; Merck) and MRS agar, respectively, supplemented with tetracycline (25 µg mL^−1^) or erythromycin (25 µg mL^−1^) (both from Merck). Plates were incubated at 32 °C (PCM) or 37 °C (MRS) for 48 h. Colonies showing semi-confluent growth were harvested, suspended in Brain Heart Infusion (BHI) broth (Merck) without antibiotics, supplemented with 25% glycerol, and stored at −80 °C.

### 4.3. Plasmid Isolation and Transformation

Plasmid DNA from antibiotic-resistant bacteria was extracted from 200 µL of their frozen cell suspensions. These suspensions were first centrifuged, washed with sterile phosphate-buffered saline (PBS; Merck), and then suspended in 200 µL of the suspension buffer supplied with the High Pure Plasmid Isolation Kit (Roche, Basel, Switzerland). Lysozyme (20 mg mL^−1^), mutanolysin (20 U) and lysostaphin (50 µg mL^−1^) (all from Sigma-Aldrich, St. Louis, CA, USA) were added to the suspensions. Cells were incubated at 37 °C for 1 h, and the plasmid DNA extracted and purified following the above kit’s protocol.

Electrocompetent cells of *L. lactis* NZ9000, *L. plantarum* NC8 and *L. casei* LB23 were prepared according to Holo and Nes [80]. Plasmid DNA (1–2 µg) was introduced into 75 µL of electrocompetent strains using a Gene Pulser apparatus (Bio-Rad, Richmond, CA, USA) following standard protocols for Gram-positive bacteria. After electroporation, cells were suspended in 1 mL of fresh media and incubated at 32 °C for 2–4 h. The cells were then plated onto appropriate media supplemented with 25 µg mL^−1^ of either tetracycline or erythromycin and incubated under the above conditions.

Plasmid DNA from antibiotic-resistant colonies was extracted and purified following the procedure of O’Sullivan and Klaenhammer [81]. To facilitate lysis, transformants were grown in GM17 (*L. lactis*) or MRS (*L. casei* and *L. plantarum*) supplemented with 40 mM of DL-threonine (Merck). Plasmids were digested with restriction endonucleases as recommended by their supplier (Takara, Saint-Germain-en-Laye, France), and the resulting profiles visualized under UV light after electrophoresis in 1% agarose gels stained with GreenSafe Premium (NZYTech, Lisboa, Portugal).

### 4.4. Minimum Inhibitory Concentrations of Antibiotics

The minimum inhibitory concentrations (MIC) of 16 antibiotics in the untransformed parental and resistant transformants were assayed by microdilution using Sensititre EULACBI1 and EULACBI2 plates (Trek Diagnostic Systems, East Grinstead, UK). Briefly, individual colonies were suspended in a sterile saline solution until reaching a density corresponding to McFarland standard 1 (≈10^8^ cfu mL^−1^). The suspension was then diluted 1000-fold in IsoSensitest (IST) broth (Oxoid, Basingstoke, UK) (for *Lactococcus*) or LSM medium (90% IST + 10% MRS) (for lactobacilli), and 100 µL of this suspension placed in each well of the Sensititre plates. These were then incubated under aerobic conditions at 32 °C for 48 h. MICs were defined as the lowest concentration at which no visible growth was observed. For some antibiotics, the concentration range of the plates was too small to determine the actual MIC; in such cases, the MICE system (Oxoid) was used following the manufacturer’s recommendations. In short, a sterile cotton swab was immersed into a sterile saline solution as above (McFarland standard 1) and used to inoculate IST or LSM agar plates. A MICE strip of the required antibiotic was placed on the plates, and incubated as above. The MIC was determined as the first concentration at which the inhibition halo contacted the antibiotic strip.

### 4.5. PCR Detection of Tetracycline and Erythromycin Resistance Determinants

The presence of tetracycline and erythromycin resistance genes was first investigated by standard PCR. Genes coding for ribosomal protection proteins conferring tetracycline resistance were targeted with the pairs of universal primers DI-DII [82] and Tet1-Tet2 [83], as well as with specific primers for *tet*(W) [84], *tet*(M), *tet*(S), and *tet*(O) [85]. Tetracycline resistance genes coding for the efflux pumps, *tet*(K) and *tet*(L), were also searched for using gene-specific primers [85]. The presence of erythromycin resistance genes was tested using specific primers for *ermA, ermB, ermC* [86], *ermF* [87], and *mefA* [88]. Amplicons were purified, sequenced, and their sequences compared to those in the NCBI database using the BLASTn tool (https://blast.ncbi.nlm.gov/Blast.cgi, accessed on 20 July 2021). The primer sequences, PCR conditions and expected amplicon sizes are listed in Appendix A.

### 4.6. Whole Plasmid DNA Sequencing, Assembly and Annotation

Purified DNA from plasmids showing different restriction patterns were sent for sequencing to Eurofins Genomics (GATC Biotech, Constance, Germany). Individual libraries were constructed using the SPRIworks Fragment Library System I Kit (Beckman Coulter, Brea, CA, USA) and pair-end sequenced (2 × 150 bp runs) in a HiSeq sequencer (Illumina, San Diego, CA, USA). Quality-filtered reads, trimmed or non-trimmed depending on the plasmid, were de novo assembled in contigs using SPAdes v3.6.2 software [89], with a k value of 127 and employing the only-assembler settings. The plasmidSPAdes algorithm [90], which uses coverage as a parameter to remove chromosomal contigs, was also used for assembly. When plasmids were not merged into a single molecule after assembly, primers were designed based on the end of the contigs, used in PCR reactions, and the sequences of the amplicons employed to join the segments. The Vector NTI program (Invitrogen, Carlsbad, CA, USA) was used to align the sequences of contigs and PCR amplicons. The same program was used for open reading frame (ORF) prediction. Deduced protein sequences larger than 50 amino acids were compared to those in the NCBI’s non-redundant protein database and manually analyzed using BLASTp.

## 5. Conclusions

In conclusion, plasmids conferring tetracycline and erythromycin resistance that were capable of replicating in LAB species were detected in Cabrales cheese-associated bacteria. For *L. lactis*, PCR and sequence analysis revealed *tet*(S) and *ermB* to be involved in tetracycline and erythromycin resistance, respectively. Non-specific plasmid-mediated tetracycline resistance in lactobacilli species was also observed. The detected plasmids were mosaic structures of modules bound by IS elements, the separate modules harboring genes coding for replication, mobilization, and antibiotic resistance functions. The detection of the same module in different plasmids suggests the existence of efficient module-exchanging mechanisms in LAB. The nucleotide identity of *tet*(S) and its flanking sequences together indicate the spread of this determinant across the cheese ecosystem to be recent. The present results reveal the potential of the procedure used in this work to detect plasmids carrying antibiotic resistance in cheese.

## Figures and Tables

**Figure 1 ijms-22-07801-f001:**
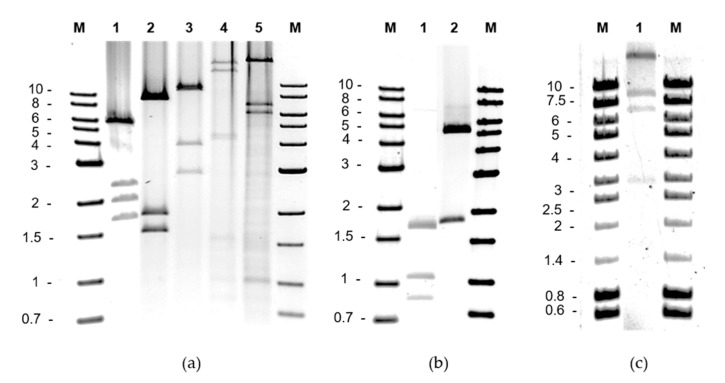
Restriction profiles of plasmid DNA obtained from transformants of tetracycline-resistant *Lactococcus lactis* NZ9000 (**a**), tetracycline-resistant *Lacticaseibacillus casei* BL23 and *Lactiplantibacillus plantarum* NC8 (**b**), and erythromycin-resistant *L. lactis* NZ9000 (**c**). Order: Panel (**a**), lanes 1 through 5, pTC1 to pTC5 digested with EcoRI; Panel (**b**), lanes 1 and 2, pTC6, and pTC7 digested with HindIII; Panel (**c**), lane 1, pERM1 digested with PvuI and EcoRI. M, Molecular weight markers.

**Figure 2 ijms-22-07801-f002:**
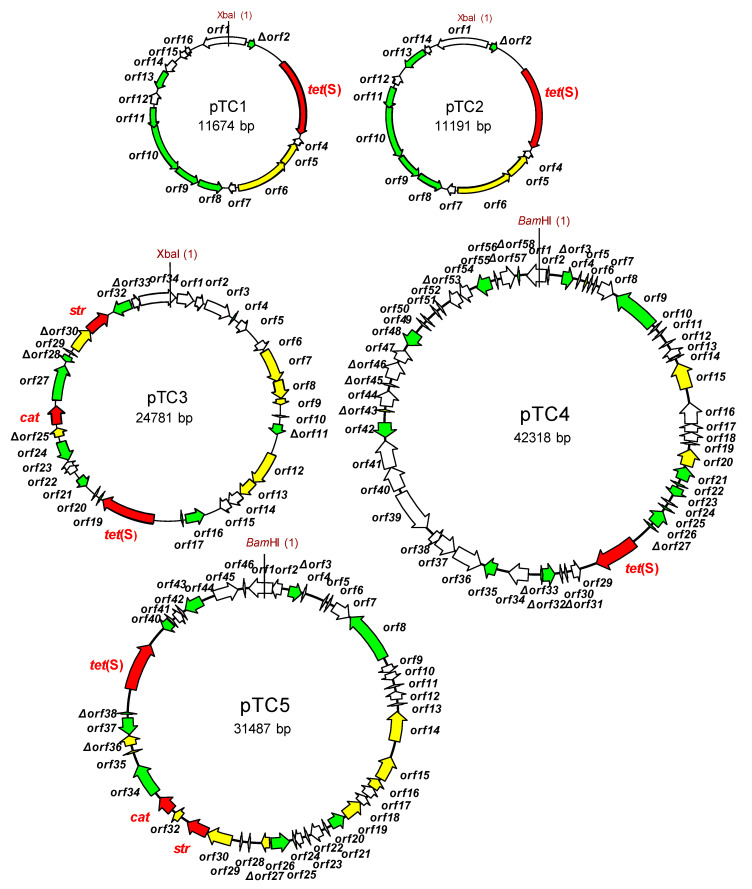
Drawn to scale genetic maps of the tetracycline resistance plasmids recovered in *Lactococcus lactis*, pTC1 through pTC5. Arrows indicate the position, direction and length of the open reading frames (ORFs) identified. The position of *tet*(S), *str* and *cat* genes are highlighted in red. Color code of the ORFs: red, antibiotic resistance genes; yellow, genes involved in replication; green, insertion sequences and genes involved in mobilization; white, genes encoding other or unknown functions.

**Figure 3 ijms-22-07801-f003:**
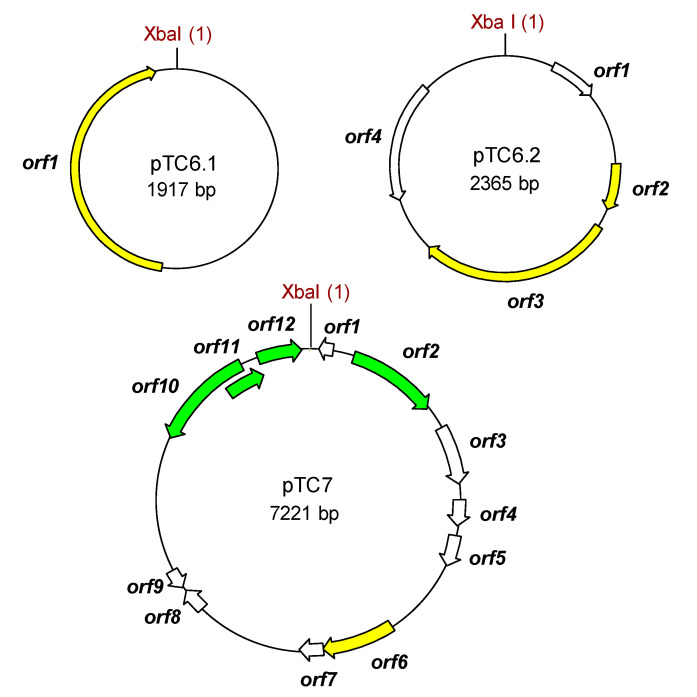
Drawn to scale genetic maps of the tetracycline resistance plasmids recovered in *Lacticaseibacillus casei* (pTC6.1, pTC6.2) and *Lactiplantibacillus plantarum* (pTC7). Plasmid features as in Figure 2.

**Figure 4 ijms-22-07801-f004:**
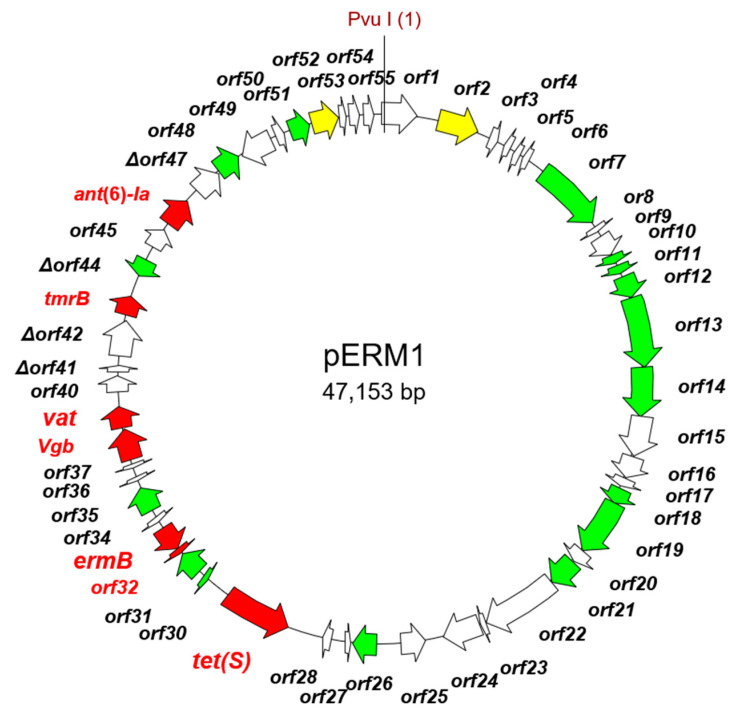
Genetic map of the erythromycin resistance plasmid pERM1 recovered in *Lactococcus lactis*. The position of *ermB* and other antibiotic resistance genes are highlighted in red. Plasmid features as in Figure 2.

**Figure 5 ijms-22-07801-f005:**
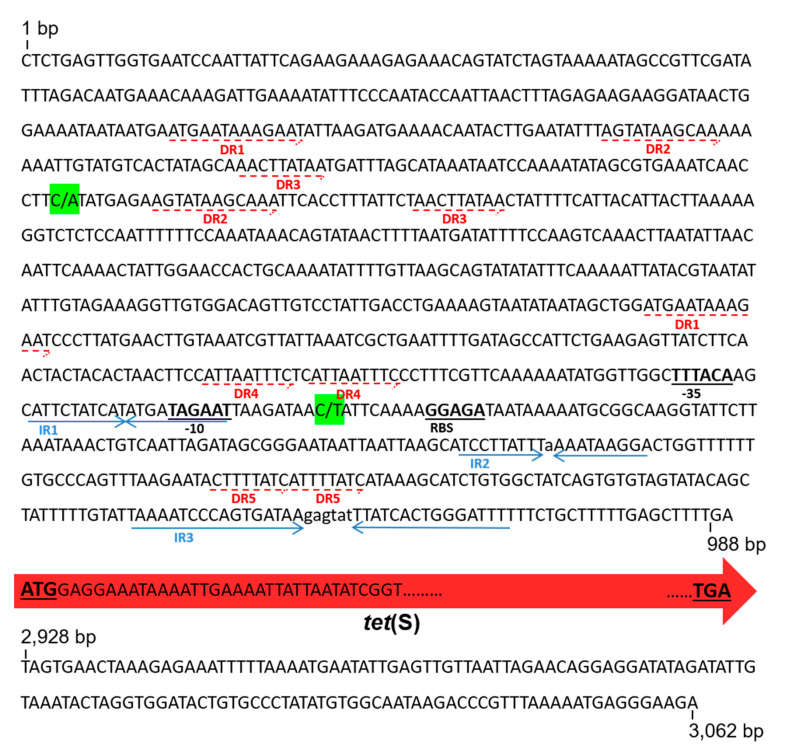
Conserved DNA sequence of the *tet*(S) gene and its surrounding regions in all resistance plasmids retrieved in *L. lactis* NZ9000. The red arrow represents the whole *tet*(S) gene from the start to the stop codons (in bold and underlined). Direct (DR) and inverted (IR) repeats upstream of *tet*(S) are underlined by solid pale-blue and dotted-red arrows, respectively. Putative -35 and -10 positions of promoter and ribosome binding site sequences are in bold and underlined. Colored in green, single nucleotide polymorphisms detected, respectively, in pTC1 (C → A) and pEMR1 (C → T).

**Table 1 ijms-22-07801-t001:** Minimum inhibitory concentration (MIC) of 16 antibiotics to the plasmid-free *Lactococcus lactis* NZ9000, *Lacticaseibacillus casei* LB23, and *Lactiplantibacillus plantarum* NC8 and their plasmid-containing transformed derivatives.

Species/Strain	Minimum Inhibitory Concentration (μg mL^−1^)															
	Gm	Km	Sm	Nm	Tc	Em	Cl	Cm	Am	Pc	Va	Q-da	Lz	Tm	Ci	Rif
*Lactococcus lactis* NZ9000	1	4	16	4	1	0.12	0.12	4	0.25	0.5	0.5	4	2	>64	16	64
*L. lactis* pTC1	1	4	16	4	128	0.06	0.12	4	0.25	0.25	0.25	4	2	>64	16	32
*L. lactis* pTC2	0.5	4	16	2	128	0.06	0.06	4	0.25	0.25	0.5	4	2	>64	16	32
*L. lactis* pTC3	0.5	4	64	2	128	0.06	0.12	32	0.25	0.25	0.5	4	1	>64	16	64
*L. lactis* pTC4	1	4	16	4	128	0.06	0.12	4	0.25	0.25	0.25	4	2	>64	16	32
*L. lactis* pTC5	1	4	32	2	192	0.12	0.12	16	0.25	0.25	0.5	4	2	>64	16	64
*L. lactis* pEMR1	1	4	>256	32	>256	>256	>256	8	0.25	0.25	0.5	>8	2	>64	16	64
*Lacticaseibacillus casei* BL23	2	16	8	4	2	0.06	0.03	4	1	0.5	>128	0.5	1	1	1	0.12
*L. casei* pTC6	1	32	16	2	32	0.25	0.06	8	0.5	1	>128	2	4	0.5	32	8
*Lactiplantibacillus plantarum* NC8	0.5	16	32	1	32	0.12	1	8	0.25	2	>128	2	4	8	32	2
*L. plantarum* pTC7	0.5	16	16	0.5	64	0.25	1	8	0.25	2	>128	1	4	16	64	2

Key of antibiotics: Gm, gentamicin; Km, kanamycin; Sm, streptomycin; Nm, neomycin; Tc, tetracycline; Em, erythromycin; Cl, clindamycin; Cm, chloramphenicol; Am, ampicillin; Pc, penicillin G; Va, vancomycin; Q-da, quinupristin-dalfopristin; Lz, linezolid; Tm, trimethoprim; Ci, ciprofloxacin; Rif, rifampicin. Grey-shaded boxes highlight the plasmid-associated increased MICs obtained in the transformants.

## Data Availability

The nucleotide sequence data of the plasmids presented in this study (pTC1-pTC7 and pEMR1) are being submitted to the NCBI database. Until they are publicly available, sequences will be provided on request from the corresponding author.

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
