# Peer review of "Directed Recovery and Molecular Characterization of Antibiotic Resistance Plasmids from Cheese Bacteria"

_ijms, 2021, doi:10.3390/ijms22157801_

Round 1
Reviewer 1 Report
Florez et al. provide an interesting technique to successfully isolate and characterize plasmids containing antimicrobial resistance genes from the cheese microenvironment. Their technique involves growing cheese bacteria in different media containing a selective pressure with antibiotics. Plasmids are purified and transformed into lactic acid bacteria, and are analyzed further by restriction fragments. Here, they isolated 5 new tetracycline-resistance plasmids and one large erythromycin-resistance plasmid. Characterization of these plasmids revealed interchangeable modules surrounded by IS elements. This suggests that there are elaborate and extensive plasmid exchange mechanisms in lactic acid bacteria.
The manuscript is well written, and the figures and tables presented the data very well. This topic will be of broad general interest to the scientific community. I have the following suggestions and comments:
Line 25: What is meant by “detach?”
Results section 2.1: Please add a rationale for each section, and explain what you are doing. For example, PCM and MRS need to be defined. Why are 3D and 60D cultures used? I know it’s described in the methods section, but it needs to be repeated here for the reading comprehension. It is just stated that “complex bands are observed” but does not mention that these were restriction digested and analyzed by gel electrophoresis. Please add more details!
Sections 2.2, 2.3 and 2.4 also needs a rationale as the opening sentences.
Line 117, briefly describe what Sensititre and MICE systems are.
In section 2.4, you should include all of the tetracycline and erythromycin genes that were screened. I know its in the methods section, but it would help the flow of the story here.
In the supplemental figures file, there is a figure on page 2, that looks like a repeat of figure S1, but labeled differently. What is this, and is it meant to be included?
Figure 2 legend, Line 215: There is no brown in these figures, and it should be “white.”
Section 4.5: Where the primer sequences also described in those references? It would be helpful to include a primer table, with appropriate references as a supplemental figure.
Minor:
Lines 31 and 35: Is “etc.” really needed here? It seems that everything is listed that would be relevant to both statements.
Line 42: Add a comma before “but.”
Line 98: Remove the semicolon, and change to a comma. Then change “these” to “which.”
Lines 118 and 397: Instead of host strains, “parental strains” would be better to use.
Line 134: Remove “and sequence analysis.”
Line 359: Define “MRS.”
Line 393: How were bands visualized? Ethidium bromide staining?
Line 400: 108 cfu should have the 8 as an exponent (108).

Author Response
First of all, we appreciated very much all comments and suggestions raised by the reviewers. Their comments and suggestions were mostly found valuable, and, as far as we are aware, all of them we have addressed, and, hopefully, conveniently assessed. Therefore, the manuscript has been amended in such a way that both readability and understandability is thought to be improved in this new/revised version; this would surely influence the impact of our work. Corrections made in the document have been highlighted in yellow, for the convenience of Editor and reviewers. In addition, the whole text, figures and tables have been reviewed and whenever a typing mistake or an error has been detected, these were corrected. English usage in the final version of the manuscript has been reviewed by Adrian Burton, an English-mother, freelance scientific translator and writer (www.physicalevidence.es; adrianburton1@yahoo.es).
Reviewer 1
Florez et al. provide an interesting technique to successfully isolate and characterize plasmids containing antimicrobial resistance genes from the cheese microenvironment. Their technique involves growing cheese bacteria in different media containing a selective pressure with antibiotics. Plasmids are purified and transformed into lactic acid bacteria, and are analyzed further by restriction fragments. Here, they isolated 5 new tetracycline-resistance plasmids and one large erythromycin-resistance plasmid. Characterization of these plasmids revealed interchangeable modules surrounded by IS elements. This suggests that there are elaborate and extensive plasmid exchange mechanisms in lactic acid bacteria. The manuscript is well written, and the figures and tables presented the data very well. This topic will be of broad general interest to the scientific community. I have the following suggestions and comments:
Line 25: What is meant by “detach?”
We use the word “detach” as a synonym of “highlight, emphasize or stand out”. To better understanding the sentence has been replace by “, and emphasize the leading role of plasmids in the spread of resistance genes via the food chain.”
Results section 2.1: Please add a rationale for each section, and explain what you are doing. For example, PCM and MRS need to be defined. Why are 3D and 60D cultures used? I know it’s described in the methods section, but it needs to be repeated here for the reading comprehension. It is just stated that “complex bands are observed” but does not mention that these were restriction digested and analyzed by gel electrophoresis. Please add more details!
Sections 2.2, 2.3 and 2.4 also needs a rationale as the opening sentences.
Line 117, briefly describe what Sensititre and MICE systems are.
The journal’s organization, in which material and methods section is located at the end of the manuscript, could difficult understandability of results if along this section the origin of data is not properly explained. For this reason and in accordance with reviewer, all the suggestions have been considered and the required explanatory information has been incorporated.
In section 2.4, you should include all of the tetracycline and erythromycin genes that were screened. I know its in the methods section, but it would help the flow of the story here.
As in previous points, supplementary data have been included to improve the flow of the manuscript.
In the supplemental figures file, there is a figure on page 2, that looks like a repeat of figure S1, but labelled differently. What is this, and is it meant to be included?
Thank you for your observation. It’s true, in Supplementary Material figure S1 is presented, but also an initial version of that. Therefore, the incomplete and repeated figure S1 has been deleted from Supplementary Material support.
Figure 2 legend, Line 215: There is no brown in these figures, and it should be “white.”
Thanks again for your observation. This is an error as consequence of working with different versions of figures during the preparation of the manuscript. The mistake has been amended.
Section 4.5: Where the primer sequences also described in those references? It would be helpful to include a primer table, with appropriate references as a supplemental figure.
Taken into account the recommendation of the reviewer, a Table (Table S2) which include the primer sequences, target gene, annealing temperature, expected amplicon size and revised references have been incorporated as supplemental data.
Minor:
Lines 31 and 35: Is “etc.” really needed here? It seems that everything is listed that would be relevant to both statements.
As recommended by the reviewer “etc” has been deleted from line 31, however, we think that in line 35 (now line 36) should be included due to the limitless environments in which the spread of antibiotic resistance can occur.
Line 42: Add a comma before “but.”
Line 98: Remove the semicolon, and change to a comma. Then change “these” to “which.”
Lines 118 and 397: Instead of host strains, “parental strains” would be better to use.
Line 134: Remove “and sequence analysis.”
Line 359: Define “MRS.”
Line 393: How were bands visualized? Ethidium bromide staining?
Line 400: 108 cfu should have the 8 as an exponent (108).
All final observations reported by the reviewer have been considered, modifying or implementing the information as required.

Reviewer 2 Report
In the work titled “Directed recovery and molecular characterization of antibiotic resistance plasmids from cheese bacteria” by Flórez et al., the authors described and proposed a methodology to prove the presence of functional tetracycline (Tet) and erythromycin (Ery) resistant genes in mobile elements (plasmid) isolated from lactic acid bacteria (LAB) of Cabrales cheese. For that, the authors isolated the whole plasmids contained in LAB isolated in PCM and MRS media in the presence of Tet and Ery and the plasmids were then transformed into three different LAB bacteria, plated on Tet or Ery added media. The growing colonies were selected and the plasmid isolated and characterized. Also, co-resistances with other antibiotics were analyzed.
I found it an interesting approach to research antibiotic-resistant mobile elements in food products, however, some issues should be addressed.
In point 2.1 and 2.2, I would like to ask the authors about how many colonies (CFU/gr of cheese) were isolated from the cheese with Ery or Tet resistance. These data should be provided. Besides and considering that in the end only 7 different Tet resistant plasmids and one Ery resistant were isolated. I would like to know if the authors consider this number realistic based on the number of colonies firstly isolated. Perhaps a single colony plasmid isolation for a selection of colonies from the first plates (the used in the whole plasmid isolation) can provide information about if the final number of plasmids detected are in the line of reality or if this number is underestimated.
Besides, I would like to ask why the authors discarded E. coli as a recipient for the functional plasmid isolation. Although they were looking for resistance mobile elements in Gram-positive bacteria, it is known that the efficiency in the transformation of the used bacteria is lower than in the case of E. coli, and Gram-positive plasmids could also replicate in Gram-negative bacteria.
From the 97 final colonies isolated from the plates, how many colonies harboured the same plasmid? which one was the most isolated? Although the authors indicated that plasmids with size between 4 and 50 Kb were isolated, the transformation of large size plasmids is less effective than small size plasmids. Since no clue about the plasmids range size used in the transformation is provided, are the authors afraid about if large size TetR and EryR plasmid are underestimated? Perhaps the author should fractions the plasmid according to size before the transformation. Another question is about the copy number of the isolated plasmid. A high copy number plasmid could be also more frequently transformed than a low copy. Did the authors consider this point? As commented before, a single colony plasmid isolation for a selection of colonies isolated initially might provide information about the plasmid variability in the sample. I’m worried if the actual results only provide a short vision of the real situation, but of course, it depends on the number of colonies TetR and EryR initially isolated. So, can the authors provide enough information about this concern?
Between lines 333 and 339, the authors explain why plasmids pTC6.1, pTC6.2 and pTC7 being negative for the presence of Tet resistant genes, provide resistance to this antibiotic. In the case of pTC7, the increment in the resistance to Tet was only 2 fold concerning the no-transformed strain, but for pTC6.1 and 1.2 is was 16 times. Can the author provide a deeper analysis to justify this based on the genes codify by these plasmids?
Author Response
First of all, we appreciated very much all comments and suggestions raised by the reviewers. Their comments and suggestions were mostly found valuable, and, as far as we are aware, all of them we have addressed, and, hopefully, conveniently assessed. Therefore, the manuscript has been amended in such a way that both readability and understandability is thought to be improved in this new/revised version; this would surely influence the impact of our work. Corrections made in the document have been highlighted in yellow, for the convenience of Editor and reviewers. In addition, the whole text, figures and tables have been reviewed and whenever a typing mistake or an error has been detected, these were corrected. English usage in the final version of the manuscript has been reviewed by Adrian Burton, an English-mother, freelance scientific translator and writer (www.physicalevidence.es; adrianburton1@yahoo.es).
Reviewer 2
In the work titled “Directed recovery and molecular characterization of antibiotic resistance plasmids from cheese bacteria” by Flórez et al., the authors described and proposed a methodology to prove the presence of functional tetracycline (Tet) and erythromycin (Ery) resistant genes in mobile elements (plasmid) isolated from lactic acid bacteria (LAB) of Cabrales cheese. For that, the authors isolated the whole plasmids contained in LAB isolated in PCM and MRS media in the presence of Tet and Ery and the plasmids were then transformed into three different LAB bacteria, plated on Tet or Ery added media. The growing colonies were selected and the plasmid isolated and characterized. Also, co-resistances with other antibiotics were analyzed.
I found it an interesting approach to research antibiotic-resistant mobile elements in food products, however, some issues should be addressed.
In point 2.1 and 2.2, I would like to ask the authors about how many colonies (CFU/gr of cheese) were isolated from the cheese with Ery or Tet resistance. These data should be provided. Besides and considering that in the end only 7 different Tet resistant plasmids and one Ery resistant were isolated. I would like to know if the authors consider this number realistic based on the number of colonies firstly isolated. Perhaps a single colony plasmid isolation for a selection of colonies from the first plates (the used in the whole plasmid isolation) can provide information about if the final number of plasmids detected are in the line of reality or if this number is underestimated.
Besides, I would like to ask why the authors discarded E. coli as a recipient for the functional plasmid isolation. Although they were looking for resistance mobile elements in Gram-positive bacteria, it is known that the efficiency in the transformation of the used bacteria is lower than in the case of E. coli, and Gram-positive plasmids could also replicate in Gram-negative bacteria.
As suggested by the reviewer, the count values (ufc/ml) enumerated on agar plates supplemented with tetracycline or erythromycin from samples on days 3 and 60 of cheese manufacture have been incorporated to the manuscript.
We think that isolation of plasmids from colonies picked directly from the first plates (the ones used in the whole plasmid isolation) does not necessary provide quicker and more accurate information on plasmid diversity, since in those plates not only can grow bacteria harbouring antibiotic resistance genes in plasmids but also in the chromosome. Therefore, the number of antibiotic-resistant colonies analysed, in this case, should be much higher than in the procedure applied in this work to properly evaluate the diversity of plasmids. Certainly, some information will be lost, particularly on that related to low copy number plasmids. However, it has been possible to characterize ARG-carrying plasmids that could represent a greater potential risk for the spread of resistance. This is in agreement to the detection of tet(S) gene as unique responsible for resistance to tetracycline in plasmids as appear pointed in this sentence “A larger copy number of the plasmids carrying tet(S) —a possibility not assessed in the present study— might help mask other tetracycline resistance genes”.
We have selected LAB as host for this work due to its abundance in dairy product, specifically, Lactococcus lactis, Lactiplantibacillus plantarum, and Lacticaseibacillus casei species. LAB plasmids cannot always replicate in E. coli, not even in all LAB species, sometimes a single species or close relatives (narrow host-range). The plasmids identified in the antibiotic-resistant lactococci were not detected among the antibiotic-resistant lactobacilli analysed in this study, and vice versa. However, we agree with the reviewer that it might be of interest to use other host species as E. coli but even other antimicrobials compounds and food systems. This option is reported in the manuscript “Broadening this approach to other antimicrobials, other dairy species, and other food systems, may help estimate the risk of transmission of plasmid-encoded antibiotic resistance via the food chain.”
From the 97 final colonies isolated from the plates, how many colonies harboured the same plasmid? which one was the most isolated? Although the authors indicated that plasmids with size between 4 and 50 Kb were isolated, the transformation of large size plasmids is less effective than small size plasmids. Since no clue about the plasmids range size used in the transformation is provided, are the authors afraid about if large size TetR and EryR plasmid are underestimated? Perhaps the author should fractions the plasmid according to size before the transformation. Another question is about the copy number of the isolated plasmid. A high copy number plasmid could be also more frequently transformed than a low copy. Did the authors consider this point? As commented before, a single colony plasmid isolation for a selection of colonies isolated initially might provide information about the plasmid variability in the sample. I’m worried if the actual results only provide a short vision of the real situation, but of course, it depends on the number of colonies TetR and EryR initially isolated. So, can the authors provide enough information about this concern?
We are in accordance with the reviewer that the transformation of large size plasmids is less effective than small size plasmids and therefore this could affect plasmid diversity analysis. As reviewer suggests, fractions of plasmids could minimize the size limitation. However, the isolation of ⁓ 42 (pTC4) and ⁓ 48 kb (pERM1) plasmids suggests that plasmid size could not be a serious limitation for the analysis if plasmids are abundant. The plasmid extraction protocol, the plasmid copy number and size, as well, the selection of host species can moderately affect the results. However, these limitations would not hinder to identify those antibiotic-resistance carrying plasmids in large abundance and therefore, as previously commented, with the greatest potential risk antibiotic resistance spreading through food chain.
Between lines 333 and 339, the authors explain why plasmids pTC6.1, pTC6.2 and pTC7 being negative for the presence of Tet resistant genes, provide resistance to this antibiotic. In the case of pTC7, the increment in the resistance to Tet was only 2 fold concerning the no-transformed strain, but for pTC6.1 and 1.2 is was 16 times. Can the author provide a deeper analysis to justify this based on the genes codify by these plasmids?
As reviewer pointed out, plasmids pTC6.1/pTC6.2 and pTC7 moderately increased MICs to tetracycline, mainly in the former. The plasmids pTC6.1/pTC6.2 have been previously identified and characterized in L. plantarum strain WCFS1 [namely, pWCFS101 (CR377165.1) and pWCFS102 (CR377164.1)], and the plasmid pTC7 in the strain L. plantarum DSM 20174 [plasmid p256 (AJ62894.1)], however no ORFs directly related to antibiotic resistance have been detected. For this reason, we suggest the presence of non-specific mechanisms such as reduced anti-biotic uptake, reduced cell permeability, the thickness and compactness of the cell wall, …. as a consequence of the ORFs in plasmids or their effect on others in the host genome. Although a further individual characterization of ORFs in plasmids could elucidate this fact, this was not the aim of the present work.
